

# Ocean alkalinity enhancement approaches and the predictability of runaway precipitation processes -

**Results of an experimental study to determine critical alkalinity ranges for safe and sustainable application scenarios**

**Niels Suitner[1], Giulia Faucher[2], Carl Lim[3], Julieta Schneider[2], Charly A. Moras[4], Ulf Riebesell[2] and Jens Hartmann[1]**

[1]Institute for Geology, Universität Hamburg, Bundesstrasse 55, 20146 Hamburg, Germany

[2]GEOMAR Helmholtz Centre for Ocean Research Kiel, Wischhofstrasse 1-3, 24148 Kiel, Germany

[3]KU Leuven, Bruges, Department of Materials Engineering, B-8200, Belgium

[4]Faculty of Science and Engineering, Southern Cross University, Lismore, NSW 2480, Australia

Correspondence: Niels Suitner (niels.suitner@uni-hamburg.de), Jens Hartmann (geo@hattes.de)

**Orcid:**

Niels Suitner:          https://orcid.org/0000-0003-3413-857X

Giulia Faucher:         https://orcid.org/0000-0001-8930-477X

Carl Lim:               https://orcid.org/0000-0002-9035-4771

Julieta Schneider:      https://orcid.org/0000-0002-7271-717X

Charly A. Moras:        https://orcid.org/0000-0001-6819-6167

Ulf Riebesell:          https://orcid.org/0000-0002-9442-452X

Jens Hartmann:          https://orcid.org/0000-0003-1878-9321



## Abstract

To ensure the safe and efficient application of Ocean Alkalinity Enhancement (OAE), it is crucial to investigate its impacts on the carbonate system. While various modeling studies showed promising results in the past, there has been a lack of empirical data to support the applicability of this technology in natural environments. Recent studies have described the effect of runaway precipitation in the context of OAE, showing that calcium carbonate formation was triggered if certain $\Omega_{aragonite}$ saturation threshold levels were exceeded. This precipitation can adversely affect the carbon storage capacity and may in some cases result in $CO_2$ emissions. Experiments at the Espeland Marine Biological Station (Bergen, Norway) were conducted to systematically study the chemical consequences of OAE deployment. The experiments lasted for 20-25 days to monitor the temporal development of carbonate chemistry parameters after alkalinity addition and the eventually triggered carbonate precipitation process. Identified uniform patterns before and during the triggered runaway process can be described by empirical functional relationships. For the $CO_2$-equilibrated approaches, total alkalinity levels (TA) of up to 6500 µeq kgsw$^{-1}$ remained stable without loss of total alkalinity (TA) for up to 20 days. Higher implemented TA levels, up to 11200 µeq kgsw$^{-1}$, triggered runaway carbonate formation. Ones triggered, the loss of alkalinity continued until $\Omega_{aragonite}$ values leveled out at 5.8-6.0, resulting still in a net gain of 3600-4850 µeq kgsw$^{-1}$ in TA. The $CO_2$-non-equilibrated approaches, however, remained only stable for TA additions of up to 1000 µeq kgsw$^{-1}$. The systematic behavior of treatments exceeding this level allows to predict the duration of transient stability and the quantity of TA loss after this period. Once triggered, the TA-loss continued in the $CO_2$-non-equilibrated approaches until $\Omega_{aragonite}$ values of 2.5–5.0 were reached, in this case resulting in a net loss of TA. To prevent a net loss of TA, treated water must be diluted below the time-dependent critical levels of TA and $\Omega_{aragonite}$ within the identified transient stability duration. Identified stability and loss patterns of added TA depend on local environmental conditions impacting the carbonate system, like salinity, temperature, biological activity, and particle abundance. Implementation of such identified stability and loss patterns into ocean biogeochemical models, capable of resolving mixing patterns of treated and untreated water parcels, would allow to predict, from the geochemical perspective, safe local application levels of TA, as well as the fate of added alkalinity, and therefore a more realistic carbon storage potential as if neglecting observed carbonate system response to OAE.



## 1 Introduction


At the current greenhouse gas emission rates, global warming well below 2°C, compared to pre-industrial levels, as targeted
by the Paris Agreement (UNFCCC, 2015) might not be achievable (Meinshausen et al., 2009; Rogelj et al., 2016). To prevent
such a development, international efforts have turned the spotlight on reducing greenhouse gas emissions globally. However,
to comply with the climate goals, greater attention needs to be paid to carbon dioxide removal (CDR) technologies. One such
marine-based technology is ocean alkalinity enhancement (OAE), a strategy that aims to chemically sequester carbon dioxide
($CO_2$) as carbonate ($CO_3^{2-}$) or bicarbonate ($HCO_3^{-}$) ions in ocean water (Kheshgi, 1995; NASEM, 2022). The concept of OAE
strives to increase the inorganic carbon storage capacity by increasing the total alkalinity (TA) of seawater (Caldeira & Rau,
2000; Hartmann et al., 2013; Köhler et al., 2010; Schuiling & Krijgsman, 2006). Given the residence time of inorganic carbon
in the ocean, it could be potentially stored for 10,000 to 100,000 years (Berner et al., 1983; Mackenzie & Garrels, 1966). An
accompanying benefit of this strategy is the parallel increase in pH, thus counteracting ocean acidification (Ilyina et al., 2013;
Köhler et al., 2010).
Tests for OAE under close-to-natural conditions are still scarce (Albright et al., 2016; Cyronak et al., 2023; Ferderer et al.,
2022; Paul et al., 2023; Sánchez et al., 2023; Yang et al., 2023). For a safe and efficient application of OAE, it is crucial to
assess the induced changes in carbonate chemistry and investigate their potential environmental impacts (Bach et al., 2019;
Riebesell et al., 2023).
In isolated case studies, prototypes for alkalinity enhancement have already been put into practice to counteract lake
acidification (e.g., Koch & Mazur, 2016; LMBV, 2017) or were discussed in context of river water alkalinity enhancement
(Sterling et al., 2023). Various application methods for OAE ranging from spreading grinded rock powder or mineral phases
(Kheshgi, 1995) to liquid addition of alkaline solutions directly into the seawater or via rivers (Hartmann et al., 2013; Sterling
et al., 2023), and electrochemical alkalinity generation (Eisaman et al., 2023; Renforth & Henderson, 2017) have been
proposed. Alkalinity enhancement could be achieved in a $CO_2$-equilibrated or non-$CO_2$-equilibrated manner, as discussed in
Schulz et al. (2023). In the non-equilibrated scenario, the seawater would gradually equilibrate over time by absorbing
atmospheric $CO_2$. The $CO_2$-equilibrated approach consists of adding alkalized water that is already in equilibrium with the
atmosphere. This means that at the point of addition, the water is put into equilibrium with the atmosphere either with
technological apparatus before release or a $CO_2$ source is used to bring the water into equilibrium after TA addition.
Alternatively, solids like $Na_2CO_3$ or $NaHCO_3$ could be used for OAE since they are already used to capture $CO_2$ from a source
(e.g., Forster 2012, 2014) before the alkaline products are disposed of.





To illustrate the impact of different alkalinity addition scenarios on various carbonate chemistry parameters, Figure 1 presents
a TA:DIC diagram modelled after Deffeyes (1965). Besides dissolved inorganic Carbon (DIC) and total alkalinity, the
Deffeyes diagram provides information on corresponding pH, $pCO_2$, and saturation state for aragonite ($\Omega_{aragonite}$).
Surpassing critical thresholds of $\Omega_{aragonite}$ saturation states for a certain period of time could result in $CaCO_3$ precipitation
(Schulz et al., 2023; Zeebe & Wolf-Gladrow, 2001). This phenomenon could lead to a runaway process, as observed in
laboratory-based experimental studies conducted by Moras et al. (2022), Hartmann et al.

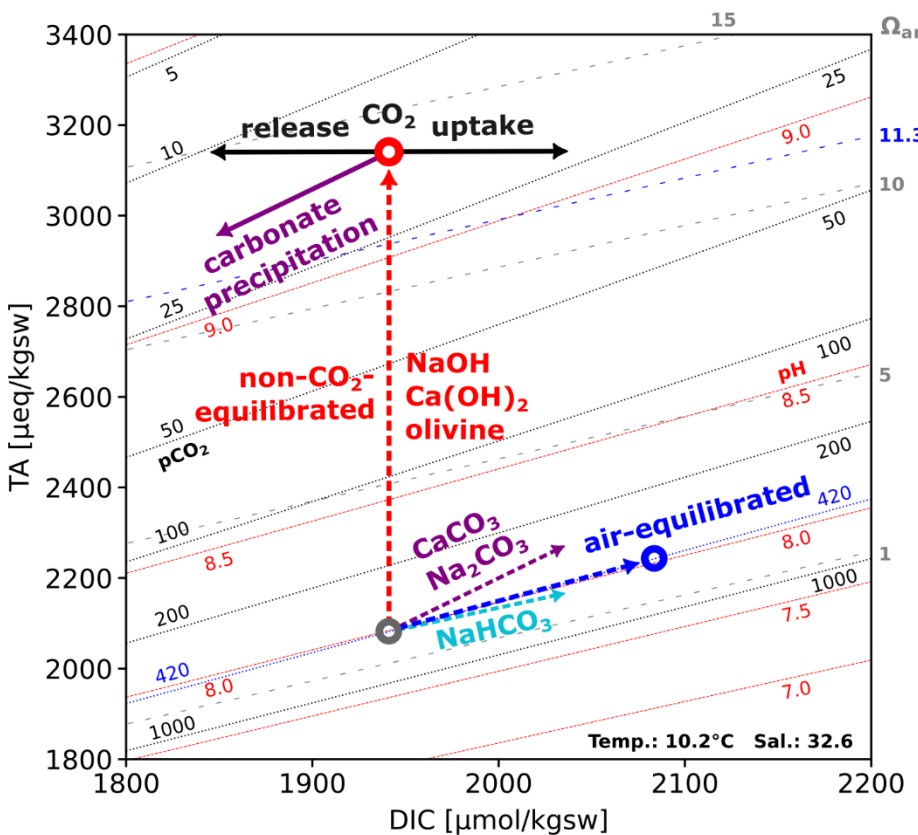

Figure 1: Example of a TA:DIC diagram after Deffeyes, 1965, in context of OAE; background contours represent iso-lines of pH (red), $\Omega_{aragonite}$ (dotted grey) and $pCO_2$ (black) values; shown contours reflect pH, $pCO_2$ and $\Omega_{aragonite}$ values for varying TA and DIC levels at given temperature and salinity conditions; dashed arrows show the impact of indicated alkalinization approaches, e.g. non-$CO_2$-equilibrated TA addition could be realized by injection or dissolution of sodium hydroxide (NaOH), an air-equilibrated TA addition (in an equilibrium to the atmosphere – $pCO_2$ 420ppm) could be achieved by utilizing a combination of $NaHCO_3$ and $Na_2CO_3$; during non-$CO_2$-equilibrated approaches the $pCO_2$ of the manipulated water is reduced, creating the potential of $CO_2$ uptake (black arrow) from the atmosphere; if the TA addition surpasses



certain critical ranges (for surface free waters under the given conditions here indicated by the blue $\Omega_{aragonite}$ contour of 11.3, calculated after Marion et al. (2009)), carbonate precipitation (purple arrow) results in the reverse changes of $CaCO_3$ dissolution

(2023), Fuhr et al. (2022), and Pan et al. (2021). This process could lead to a net loss in $CO_2$-storage potential and will result
in a leakage of TA and DIC.
Considering the complexity and variety of possible environmental impacts of OAE scenarios, systematic empirical
investigations of alkalinization approaches seem to be vital providing meaningful sustainability assessments. This study aims
to assess the geochemical impacts of alkalinity addition in seawater by refining and improving upon the experimental setup of
Hartmann et al. (2023), testing $CO_2$-equilibrated and non-equilibrated TA enhancement scenarios in natural seawater.
Incubation experiments were conducted with extended TA ranges and runtimes, along with increased sampling frequency and
enhanced resolution of the TA gradients. Experiments were designed to identify stability ranges of the added alkalinity and
characterize critical thresholds that trigger the runaway precipitation process.



## 2 Methods

### 2.1 Experimental setup

Four sets of experiments were conducted between May and July 2022 using natural seawater from the Raunefjord (60.27°N, 5.20°E) close to the Espeland Marine Biological Station (Bergen, Norway). All four experiments used the same setup. 250 ml polystyrene cell culture bottles were filled with filtered seawater in a flow-through incubation box (PMMA) and incubated outdoors to follow the local light conditions. The box was covered in blue foil (172 Lagoon Blue foil, Lee filters, Burbank, CA, United States) to mimic the light conditions in the fjord at a depth of ~5 m. The temperature was regulated by recirculating fjord water in the incubation box, thus ensuring that the incubation temperature matched that of the fjord. To prevent the occurrence of substantial headspace throughout the experiment, each treatment level was divided into 3-4 separate bottles. The division allowed for progressive volume removal during sampling while reducing the potential for gas exchange processes. Within each experiment, a new set of bottles was opened sequentially after 3-4 samplings. Alkalinity was enhanced using a 0.5 M NaOH (sodium hydroxide) stock solution for the non-$CO_2$-equilibrated and a combination of $NaHCO_3$ (sodium bicarbonate, 0.4 M) and $Na_2CO_3$ (sodium carbonate, 0.2 M) stock solutions for the preparation of the $CO_2$-equilibrated treatments. The latter were adjusted to attain equilibrium with the surrounding air's $CO_2$ concentration (~420 ppm). For each of the two, the experimental setups encompassed: 1. abiotic conditions, achieved by removing organisms via filtration through a 0.2 µm filter, 2. biotic where only the small phytoplankton community was included by using a 50 µm filter mesh to remove larger particles and organisms. An overview of the reached TA-levels and step sizes, runtimes and temperature ranges is given in Table 1. The experiments were conducted over a span of two months, with each incubation run for 20 or 25 days. The experiments were therefore partially temporally separated, resulting in slight variations in starting conditions and average temperatures, ranging from 10 to 16°C. Biotic and abiotic treatments were simultaneously conducted within each equilibration mode. The initial carbonate chemistry parameters of the collected seawater before manipulation were relatively constant for all approaches ($TA_{initial}$ ~2190 ± 5 µeq kgsw$^{-1}$, DIC ~1890 ± 5 µmol kgsw$^{-1}$, pH ~8.2 ± 0.02, Sal. ~32.6 ± 0.1).




Table 1: Overview experimental design Bergen 2022

| # | Seawater conditions | $CO_2$ state to atmosphere | Alkaline material | Runtime [days] | Range $TA_{added}$ [µeq kgsw$^{-1}$] | $TA_{added}$ gradient steps [µeq kgsw$^{-1}$] | Temperature [°C] |
|---|---|---|---|---|---|---|---|
| I | biotic | air-equilibrated | $Na_2CO_3$/ $NaHCO_3$ | 20 | 0-2800 | 200 | 12-15 |
| | abiotic | | | 20 | 0-9200 | 200/800 | 12-16 |
| II | biotic | non-equilibrated | NaOH | 25 | 0-2800 | 200 | 10-11 |
| | abiotic | | | 25 | 0-3400 | 200 | 11-13 |

### 2.2 Sampling and measurements

For carbonate chemistry analysis, 40-50 ml of incubated water were taken per sampling day and treatment level. Using a
peristaltic pump connected to a 0.2 µm syringe filter, samples were filtered immediately to stop further reactions, remove
particles and prepare each sample for measurements. All treatments were measured without replicates for TA, pH, temperature
and salinity, and biotic treatments were further analysed to assess the biological responses. An accompanying publication is
going to describe the impact of enhanced alkalinity on the included phytoplanktonic communities during the first 6 days of the
biotic incubation experiments. A selection of filtrates was saved for scanning electron microscopy (SEM) analysis. Minor
shifts in pH, DIC and $\Omega_{aragonite}$ originate from increasing water temperatures during the runtimes of the experiments,
photosynthetic activity in the biotic approaches or minor ingassing from the headspace of the reactor bottles.
Methods and devices for measuring TA, pH, temperature, and salinity were identical to experiments I and II from Hartmann
et al. (2023). Total alkalinity was determined by titration with a 0.02M hydrochloric acid, using an 888 Titrando autosampler
(Metrohm). TA measurements were corrected against certified reference materials (CRM batch 193), supplied by Prof. Andrew
G. Dickson laboratory, Scripps Institution of Oceanography (USA). A WTW multimeter (MultiLine® Multi 3630 IDS) was
used to measure pH (SenTix 940 pH-electrode), temperature and salinity (TetraCon 925 cell, Xylem). The pH-probe was
calibrated with WTW buffer solutions according to NIST/PTB in four steps (1.679–9.180 at 25 °C) and corrected for seawater
after Badocco et al. (2021). DIC, $pCO_2$ and saturation states were calculated using CO2SYS Excel version 2.5 (Pierrot et al.,
2006), including error propagations based on Orr et al. (2018). Constants in CO2SYS were set to Lueker et al. (2000) for $K_1$
and $K_2$, Dickson (1990) for $KHSO_4$, and Perez and Fraga (1987) for KHF and Lee et al. (2010) for $[B]_T$ Value, and the pH was
calculated on the total scale.





The physical appearance of precipitated particles and their elemental composition were analyzed by two separate SEM setups:
1. Tabletop Microscope Hitachi TM4000plus (University of Hamburg) and 2. Zeiss Gemini Ultra55 Plus at (CAU Kiel), both
equipped with an energy-dispersive X-ray spectroscopy (EDX) detector.

**2.3 ΔTA equation**
Based on the concept from Hartmann et al. (2023), the subsequent equation is utilized to simplify the characterization of
reached values or alterations in TA:

$$\Delta TA_{net} = TA_{final} - TA_{initial} = \Delta TA_{added} + \Delta TA_{loss}$$

$\Delta TA_{net}$: net change of TA

$TA_{final}$: absolute reached TA after TA addition (measured)

$TA_{initial}$: initial TA of used seawater (measured)

$\Delta TA_{added}$: amount of increased TA by alkalinity addition

$\Delta TA_{loss}$: amount of TA decline during the experiment (negative sign)



# 149   3 Results

## 150   3.1 CO₂ equilibrated experiments





Figure 2: Temporal TA, $\Omega_{aragonite}$ and pH evolution in $CO_2$-equilibrated experiments, under biotic (left) and abiotic (right) conditions; each diagram represents a specific carbonate chemistry parameter investigated, i.e., measured TA (a. and d.), $\Omega_{aragonite}$ (b. and e.) and pH (c. and f.) as a function of added TA and time point; legends for sampling days are given in c. (biotic) and f. (abiotic); initial conditions of the used seawater before manipulation: $TA_{initial}$ ~2190 µeq $kgsw^{-1}$, $\Omega_{aragonite}$ ~2.5-3.0, pH ~8.2, Sal. ~32.6, biotic: Temp. 12-15°C, abiotic: 12-16°C; be aware of the differing scales on the y-axes for each show parameter


In the biotic $CO_2$-equilibrated experiment an air-equilibrated alkalinization of up to 2800 µeq $kgsw^{-1}$ could be achieved during
the 20-day runtime. All carbonate chemistry parameters remained constant, showing that a TA addition slightly above the
estimated critical $\Omega_{aragonite}$ value for pseudo homogenous precipitation of 11.3 (after Marion et al., 2009) could be achieved.
This level was maintained for 20 days without causing any $CaCO_3$ precipitation, as illustrated in Fig. 2a-c.
By extending the alkalinity range up to 9200 µeq $kgsw^{-1}$ in the abiotic air-equilibrated experiment $\Omega_{aragonite}$ values far above
critical levels were reached, resulting in extensive carbonate precipitation in a runaway style (Fig. 2d-f). All targeted alkalinity
levels were achieved ~3min after alkalinity addition (day 0), and a decline in TA was observed in treatments above $\Delta$TA 3600
(corresponding to $\Omega_{aragonite}$ of 14.6). Starting with the highest treatment levels after 1 day, precipitation was triggered in all
batches with a $\Delta$TA above 3600 over the runtime of 20 days (see Fig. 2d). Following precipitation, once $\Omega_{aragonite}$ reached
values of 5.8-6.0, the process halted, resulting in a linear alignment of final TA values. The TA loss rate was significantly
reduced towards the end of the precipitation procedure; however, it could not be excluded that the process would have
continued if the experiment had proceeded. Despite substantial total alkalinity loss attributed to runaway precipitation, all
treatments involving secondary mineral formation still achieved a net gain in TA ranging from 3600 to 4850 µeq $kgsw^{-1}$.
Nevertheless, pH values in batches which underwent the precipitation process were accompanied by an acidification below
the initial seawater pH level.





**3.2 Non-CO₂-equilibrated experiments**



Figure 3: Temporal TA, $\Omega_{aragonite}$ and pH evolution of non-$CO_2$-equilibrated experiments, biotic (left) and abiotic (right);

each graph represents a specific sampling day; initial conditions of the used seawater before manipulation: $TA_{initial}$ ~2190

$\mu eq\ kgsw^{-1}$, $\Omega_{aragonite}$ ~2.5-3.0, pH ~8.2, Sal. ~32.6, biotic: Temp. ~10-11°C, abiotic: Temp. ~11-13°C


Alkalinity enhancement in the biotic and abiotic non-$CO_2$-equilibrated experiments achieved a steady increase up to an addition
of 2400 $\mu eq\ kgsw^{-1}$ at day 0 (Fig. 3). Under the given local conditions (Temp. 10-11°C, Sal. 32.6), exceeding the $TA_{target}$ level
of 4570 $\mu eq\ kgsw^{-1}$ led to a drop back to 4450 ± 60 $\mu eq\ kgsw^{-1}$ after ~3min, irrespective of the quantity of further added
alkalinity. For the biotic non-$CO_2$-equilibrated approach (Fig. 3a-c) treatments from $\Delta TA$ 1400 to 2800 showed a decrease in
alkalinity during the subsequent 25-day runtime, as a consequence of secondary mineral formation. The precipitation process
uniformly came to halt in a range of 1230 ± 60 $\mu eq\ kgsw^{-1}$, corresponding to an $\Omega_{aragonite}$ of 4-5. A similar behavior was
observed in the abiotic non-$CO_2$-equilibrated experiment (Fig. 3d-f). Slightly higher water temperatures (11-13°C) in
comparison to the biotic approach (10-11°C), potentially led to an earlier decline of alkalinity and lower final in TA (1030 ±
60 $\mu eq\ kgsw^{-1}$) and $\Omega_{aragonite}$ (2.5-4). All treatment levels from $\Delta TA$ 1200 to 3400 showed precipitation during the 25-day
runtime. Unlike the abiotic $CO_2$-equilibrated approach, the runaway precipitation in both non-equilibrated experiments resulted
in a net-loss of alkalinity, while pH values remained in a range of 9.0-10.1. Nevertheless, despite relatively high $\Omega_{aragonite}$ values
of up to ~17 (biotic) and ~15 (abiotic) in the non-$CO_2$-equilibrated experiments, after 25 days the alkalinity was still constant
in all treatments below $\Delta TA$ 1200.
**3.3 TA:DIC diagrams**
TA:DIC diagrams in Fig. 4a-e provide a contextualized overview of trend and temporal development of carbonate chemistry
parameters. Simultaneous TA and DIC enhancement in the $CO_2$-equilibrated experiments led to the characteristic diagonal
gradient, while non-$CO_2$-equilibrated approaches resulted in a straight vertical increase in TA as a consequence of $OH^-$
injection (also see Fig. 1). Treatments exhibiting secondary carbonate formation followed the strict 2:1-$\Delta TA$:$\Delta DIC$ decline
ratio during the precipitation phase, leading to a consistent alignment of data points in straight trend trajectories. The tendency
for consistent linear trends in TA/DIC during alkalinization and precipitation processes in the conducted experiments, allows
to visually trace the origin of shapes and temporal development trends of pH and $\Omega_{aragonite}$ in Figures 2 and 3 by utilizing
exhibited TA:DIC diagrams. For example, the consistency of $\Omega_{aragonite}$ values within treatments that underwent the runaway
process allows to predict the final state of other carbonate chemistry parameters, oriented on the shape and position of the
related $\Omega_{aragonite}$ contour (see Fig. 4b-e). The immediate drop back to 4450 ± 60 $\mu eq\ kgsw^{-1}$ in both non-$CO_2$-equilibrated
experiments showed a consistent pattern in dislocation of target and measured TA and DIC values, following a steady declining
$\Delta TA$:$\Delta DIC$ loss ratio of 4.9-2.3 from highest to lowest target alkalinity levels (Fig. 4f), indicating the formation of non-
carbonate-bearing secondary phases, as $Mg(OH)_2$ (also see section 4.3).











Figure 4: TA:DIC diagrams (a.) $CO_2$-equilibrated (biotic), (b.) $CO_2$-equilibrated (abiotic), (c.) zoom into precipitating treatments in (b.), (d.) non-$CO_2$-equilibrated (biotic), (e.) non-$CO_2$-equilibrated (abiotic), a selection of data points is labeled with their corresponding sampling day; (f.) comparison of targeted and measured values for treatments with immediate precipitation (blue) abiotic (red) biotic non-$CO_2$-equilibrated, indicating the formation of a non-carbonate phase like $Mg(OH)_2$; note that due to varying temperatures during the experiments, given the $\Omega_{aragonite}$ and pH contours in all diagrams are temperature and salinity-dependent, potentially resulting in slight inaccuracies in showing exact values for individual data points

## 3.4 TA loss rates

TA-loss rates for treatments which underwent the precipitation process exhibited similar relationships, independent of the $CO_2$-equilibration state. In regular patterns, elevated initial TA-levels induced an earlier initiation of the exponential decay process accompanied by increased TA loss rates within each experiment (see Figs. 5 and S4). Absolute rates were dependent on the potential for TA loss within each treatment, regulated by the initial DIC and $\Omega_{aragonite}$ values. Irrespective of the initial TA-level, treatments that showed an immediate precipitation in both non-$CO_2$-equilibrated experiments exhibited almost identical development trends during the precipitation process. Figure 6 showcases the related temporal TA development of the abiotic $CO_2$-equilibrated and biotic non-$CO_2$-equilibrated approaches. Outlier values from the few sampling days were excluded for the calculation of TA loss rates in the abiotic CO2-equilibrated and abiotic non-CO2-equilibrated experiments. For details see section "outliers" in the supplements.




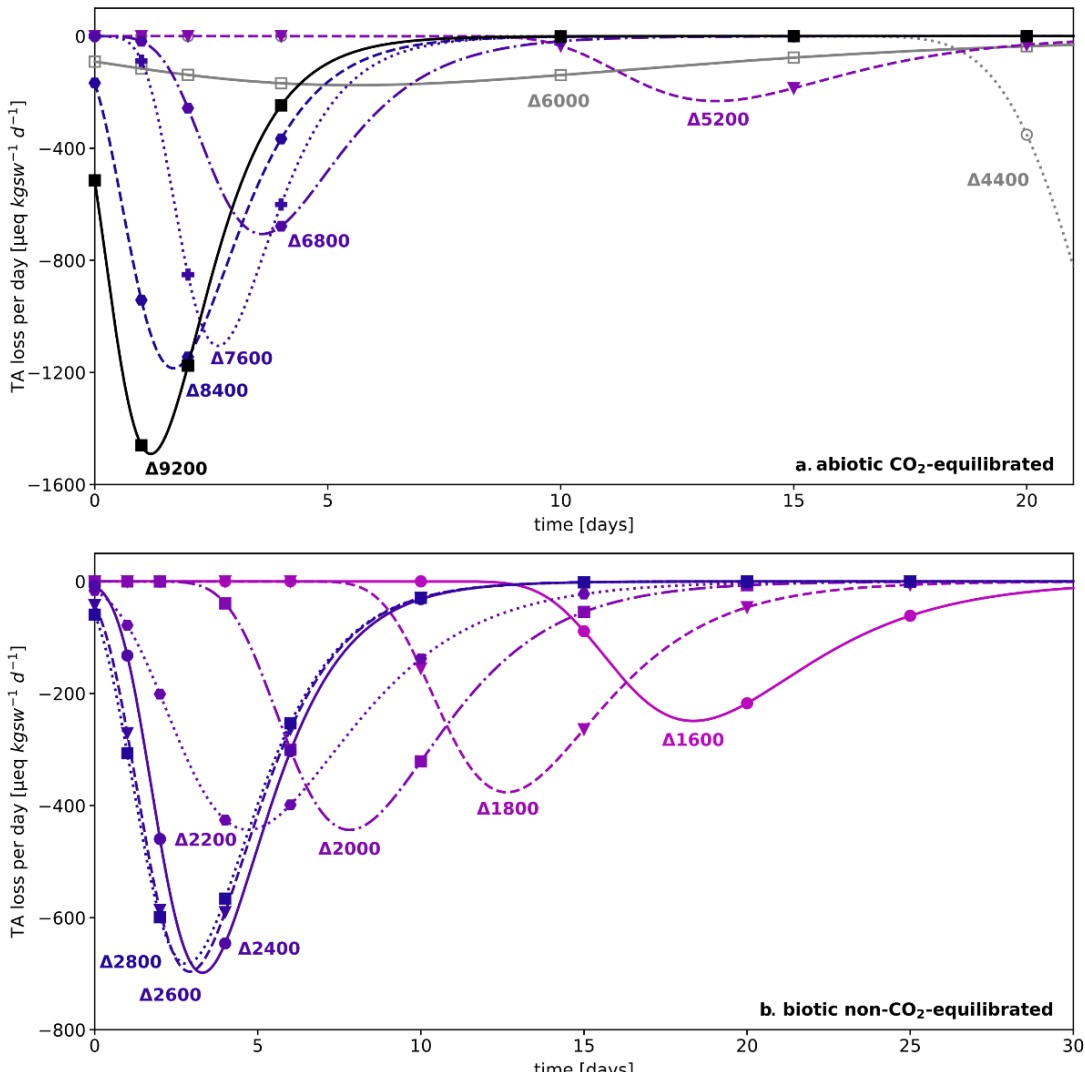

Figure 5: TA loss rates per day in (a.) abiotic $CO_2$-equilibrated and (b.) biotic non-$CO_2$-equilibrated experiments showing precipitation processes, rates were calculated based on differentiating functions determined by a sigmoidal curve fit model of the temporal development of TA (see Fig. 6); due to missing data points, rates for treatment levels Δ4400 and Δ6000 in (a.) could not be determined. Regarding ΔTA6000 see description of outliers in the supplements; for TA loss rates of the abiotic non-$CO_2$-equilibrated experiment see Fig. S4





Figure 6: Temporal development of TA in (a.) abiotic CO$_2$-equilibrated approaches and (b.) biotic non-CO$_2$-equilibrated experiments; compare to related TA loss rates in Fig. 5; see Fig. S2 and S3 for the biotic CO$_2$-equilibrated and the abiotic non-CO$_2$-equilibrated approaches





**3.5 SEM**

SEM images of the filtered residua of the experiments show a variety of common shapes of aragonite precipitates. Throughout all conducted experiments in this study visual identifiable precipitates only appeared in treatments which also exhibited a decline in TA. The morphologies of the particles in the $CO_2$-equilibrated and non-$CO_2$-equilibrated treatments were identical if secondary mineral formation was triggered. The quantity, structures and shapes of the particles evolved with increased alkalinity. Figure 7 provides examples of different development stages over the runtime of 6 days in the non-$CO_2$-equilibrated biotic experiment. The bulk of particles showed central stems, which branched out to each end. Morse et al. (2007) described the more developed shapes as "broccoli" structures, due to its physical appearance, while Nielsen et al. (2014) entitled the less branched shapes as "sheaf-of-wheat" bundle. These symmetric particles were the dominant appearing shapes of secondary phases. Treatment levels with initial precipitation showed early stages of stem-like structures with no or very little branching. With higher alkalinity addition more advanced shapes and sizes were predominant, characterized by progressive outbranching. Most developed stages exhibited a merging of the fanned out ends to form closed spheres. Next to the dominant simple stems, multipolar particles, with up to six branches were observed, at all development stages. Despite the variable initial branching numbers, the growth behavior followed the same patterns. Following the scheme presented in Fig. 7 all variants finally reached a closed structure. Observed ellipsoid shaped particles might indicate that the previous precipitate was bi-polar, while more spherical ones had a multi-polar origin. No indications for hollow stems, like described in Hartmann et al. (2023), could be observed. Sizes vary from 2-5 µm for initial shapes, to 10-30 µm for non-closed "broccoli" particles and up to 80 µm for complete spherical forms. For an overview of occurrence and distribution of particle sizes and shapes see Fig. 8(a-f). Consistently higher loss of TA during the runaway process resulted in greater numbers and more developed stages in the precipitates. EDX-analysis uniformly identified the precipitates as Ca-dominated carbonates (Ca: 7.8-12.3 mol%, Mg: 2.1-5.3 mol%, C: 13.7-25.8 mol%, O: 61.0-66.2 mol% - Zeiss Gemini Ultra55 Plus (CAU)) with indications of a relatively high content of Mg carbonates phases.



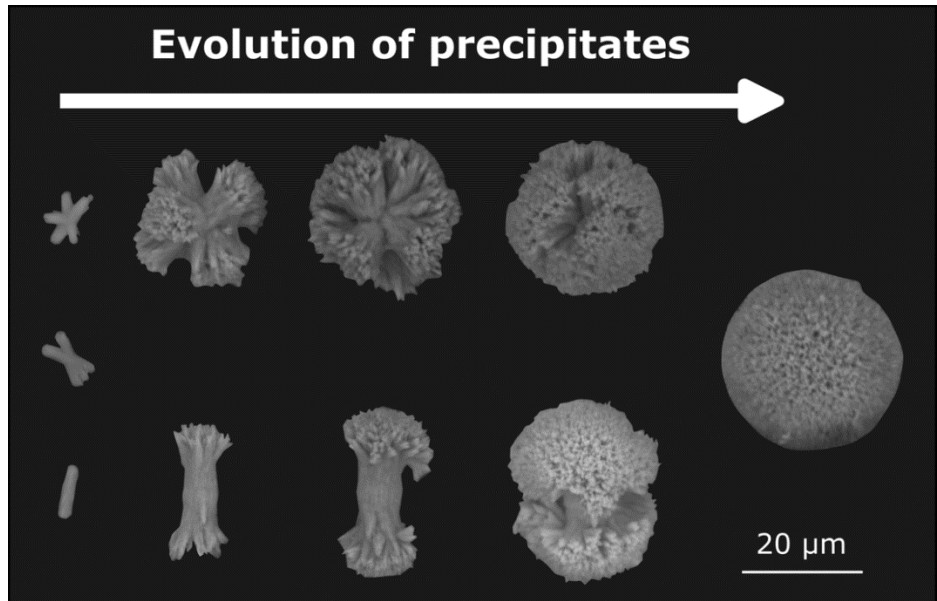

Figure 7: Scheme of the evolution of precipitates, showing a selection of precipitated particles in different development stages. Growth of the initial "stem" structure is accompanied by outbranching on each end. Independent of the polarity of the initial stems, developed particles uniformly form spherical shapes, Tabletop Microscope Hitachi TM4000plus (UHH)



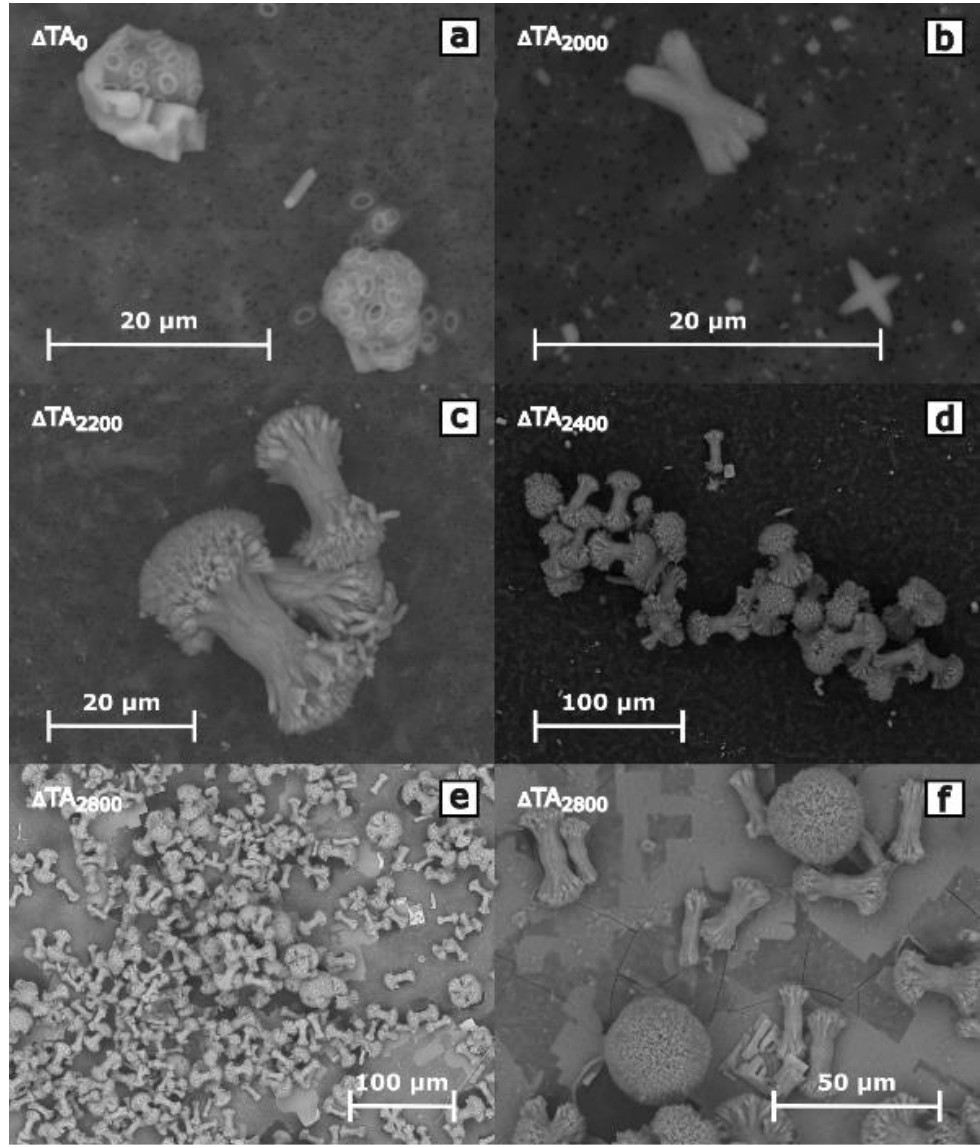

Figure 8: SEM images, overview of shapes and occurrences of precipitates in the biotic non-CO$_2$-equilibrated treatments; no difference could be determined in shapes and appearance within other experiments; compare to Boon et al. (2020), Morse et al. (1997), Pan et al. (2022), Nielsen et al. (2014) and Hartmann et al. (2023), who showed similar shaped carbonate/aragonite precipitates, Tabletop Microscope Hitachi TM4000plus (UHH)



**4 Discussion**

The stability of achieved TA enhancements varied from several hours to weeks, depending primarily on the resulting $\Omega_{aragonite}$, the $CO_2$-equilibration state, local environmental conditions, and the quantity of introduced alkalinity. While target TA levels were achieved within acceptable tolerances, treatment levels exceeding pH values of approximately 10.3 failed to achieve the intended TA values when measured within three minutes after application. Such observation is potentially the result of immediate magnesium hydroxide formation, buffering the injected alkalinity, as indicated by Eisaman et al. (2023) and Cyronak et al. (2023). While runaway calcium carbonate formation was demonstrated in previous research (Moras et al., 2022; Hartmann et al., 2023), a systematic description of TA loss with respect to time and saturation state could be established here. This allows for the prediction of TA loss behavior when local environmental parameters are well-defined. Consequently, such systematic studies will provide needed parameterized functions for models to assess the consequences of OAE before application (Fennel et al., 2023). Together with addressing the mixing of treated and untreated water, ultimately diluting the additional seawater TA, an assessment of the stability of alkalinity could be generated, if sufficient systematic studies were conducted. Remarkably in the $CO_2$-equilibrated approach for additions of up to 3600 µmol kgsw$^{-1}$, no TA loss was observed within the first 20 days, highlighting the relevance of the equilibration state of the carbonate system for the stability of alkalinity.

**4.1 Runaway CaCO$_3$ precipitation**

While the objective of this study was to detect stable alkalinity ranges, exceeding critical limits caused runaway carbonate formation, which leveled out at a new equilibrium. EDX-analysis of the precipitates (see section 3.5) and the 2:1 ΔTA:ΔDIC decline ratios (Fig. 4) confirmed the formation of CaCO$_3$ phases when pH-values were below 10.3.

Independent of the $CO_2$-equilibration state or initial treatment level, the temporal TA development patterns after the start of runaway precipitation could be fitted with a sigmoidal function. Start of runaway precipitation, TA loss rate, and duration of TA decline (Figs. 2-6) varied with temperature and initial TA and DIC treatment levels, but followed a general pattern (see Fig. 9):

    **1.** nucleation phase – stage of generation or provision of sufficient surface area to trigger the runaway process

    **2.** precipitation phase – stage of exponential decay in TA and DIC in a 2:1 ratio, due to the runaway process, until the
    potential declines significantly with reduced $\Omega$aragonite values

    **3.** new equilibrium - final state after the runaway process ended where the changes in TA and DIC might be too low
    to be measured



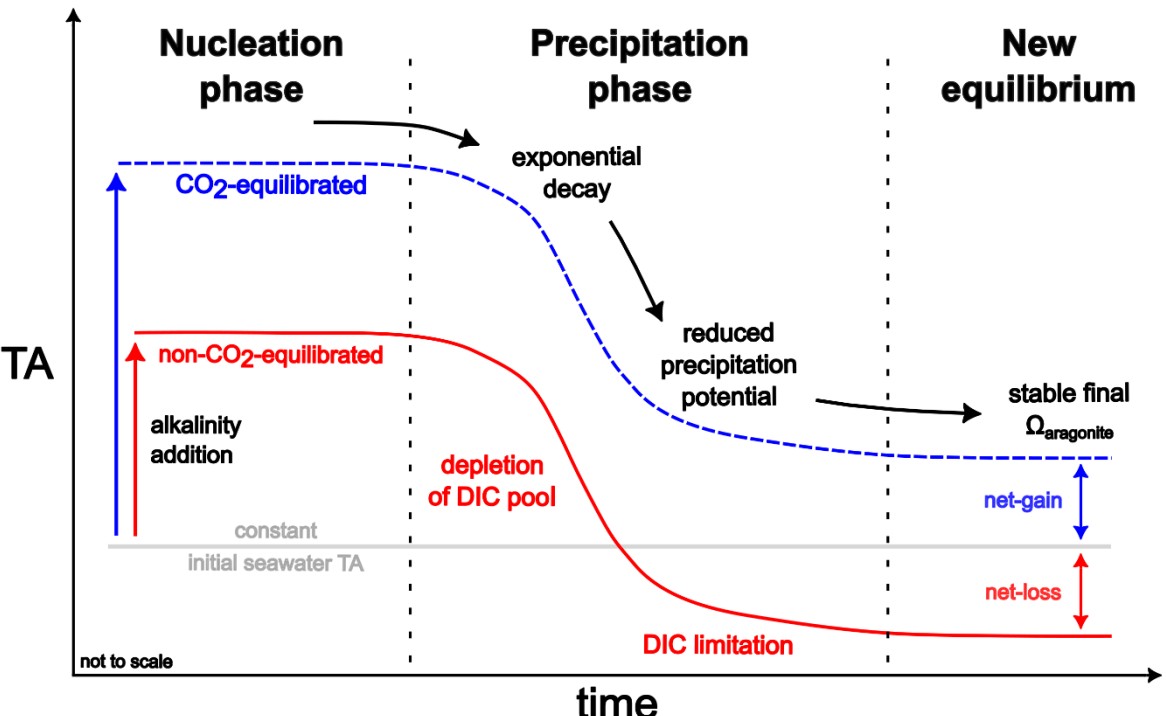

Figure 9: Concept of carbonate runaway precipitation, showing the generalized evolution of TA for non-$CO_2$-equilibrated and $CO_2$-equilibrated alkalinity addition scenarios, deduced from experimental results of this study (not to scale). As nucleation is a time dependent process, despite overcritical $\Omega_{aragonite}$ values, a stable temporal state without observable precipitation exists, depending on the physicochemical conditions ranging from seconds to years. While the secondary carbonate formation in the non-$CO_2$-equilibrated treatments result in TA values below the initial seawater levels, $CO_2$-equilibrated treatments might, despite a substantial TA loss during the runaway process, achieve a net gain in alkalinity. For further descriptions see text.

### 4.1.1 Nucleation phase

The duration of the nucleation phase varies depending on the quantity and form of added alkalinity, and the alterations in the saturation state, in this work ranging from immediate precipitation to several weeks. However, the nucleation phase might last as long as thousands of years (c.f., Pytkowicz, 1973). To date, only a small amount of data is available to parameterize the duration of this phase systematically. These data are needed for models to assess the consequences of OAE applications (Fennel et al., 2023).

Other factors such as temperature or the presence of suitable surfaces for pseudo-homogeneous or heterogeneous precipitation that were not studied here have an influence on the duration of the nucleation phase and have been suggested as triggers for



CaCO$_3$ precipitation. These suitable surfaces can include but are not limited to fluvial or marine re-suspended particles (Wurgaft et al., 2021; 2016), seafloor sediments (e.g. CaCO$_3$, quartz particles; Moras et al., 2022), small biotic and abiotic particles (< 50 µm) (Hartmann et al., 2023), seagrass, shells, biofilms, and biological activity (Aloisi et al., 2006, Zhu & Dittrich, 2016).

Dissolving alkaline particles like Ca(OH)$_2$ or Mg(OH)$_2$ for OAE could also serve as starting points for carbonate formation (Moras et al., 2022; Hartmann et al. 2023). When relying on solid alkaline materials, Moras et al. (2022) and Schulz et al. (2023) suggested that an $\Omega_{aragonite}$ of 5 should not be exceeded, above which CaCO$_3$ runaway precipitation appears to be triggered. However, under the conditions of this study, i.e., with liquid alkaline material, the $\Omega_{aragonite}$ threshold for the initiation of spontaneous pseudo-homogeneous carbonate formation in particle-free seawater is approximately 11.3 at a salinity of 32.6 and ~11°C (Marion et al., 2009 – based on data from Morse & He, 1993). Most ocean surface waters are naturally oversaturated with carbonates ($\Omega_{aragonite}$ ~2-5, Olsen et al., 2018), yet no obvious spontaneous inorganic carbonate formation is occurring., as the presence of Mg$^{2+}$ (Berner, 1975; Pan et al., 2021), phosphate (Burton & Walter, 1990), and dissolved organic matter species (Chave & Suess, 1970; Kellock et al., 2022) are known to delay or inhibit precipitation of CaCO$_3$. Since Mg$^{2+}$ in an open ocean context correlates to salinity, its concentration could vary depending on the local conditions (Moras et al., 2023), while phosphate and DOM concentrations are related to biological processes and seasonal changes.

### 4.1.2 Runaway precipitation phase: general patterns

The precipitation phase, characterized by the previously discussed parameters guiding the runaway patterns, might also be influenced by the concentration and quality of formed particles. In contrast to a natural open ocean environment, where precipitates could sink and be removed from the alkalinity-enhanced water, the experimental setup here did not take this into account.

It is imperative to investigate if the particle export mechanism could affect the shape of the identified runaway precipitation patterns, e.g. by lower TA loss rates due to less available suitable surface areas for carbonate formation. The experiments in this study were performed in bottles, where the presence of precipitation became evident through a fine whitish coating forming on the inner surface of the water-exposed part of the bottles. Despite being a laboratory artefact, the abundant presence of suspended particles suggests that in the open ocean, similar precipitation patterns could occur. The observation of crystal growth on the bottle walls suggests that the results here and the functional relationships of the runaway precipitation might be impacted by the experimental setup, leading to higher precipitation rates due to increased potential for TA loss. Therefore, field experiments addressing this issue and confirming or improving the parameterization of the loss functions are recommended.



In natural settings, comparable TA decline patterns were observed in river plumes with high degrees of suspended particles
(Wurgaft et al., 2021; 2016) or whiting events on the Great Bahama Bank (Broecker & Takahashi, 1966; Morse et al., 2003).
One study suggested that with thermohaline stratification and moderate background saturation states in an open water column,
TA loss due to carbonate formation may happen because of strong evaporation of water in the eastern Mediterranean Sea
(Bialik, 2022). The observation that runaway events could occur naturally under certain constrained conditions highlights the
importance of identifying underlying processes before OAE applications are implemented, as the higher saturation states
induced by OAE could make such events likelier.
While the fundamental patterns of changes in the carbonate system parameters during the runaway process were dictated by
carbonate formation, the starting and ending points of the procedure were dependent on the initial TA/DIC configuration and
the resulting $\Omega_{aragonite}$ achieved through manipulation. The observed differences in TA-loss in $CO_2$-equilibrated and non-$CO_2$-
equilibrated approaches were therefore expected. Under well-defined circumstances and aware of a practical final $\Omega_{aragonite}$
saturation state range, the consequences of a completed runaway precipitation process should, in theory, therefore be
predictable.
As shown in Figures 2-4, treatments which underwent a runaway process approached relatively uniform final $\Omega_{aragonite}$ values,
indicating that $\Omega$-values served as the decisive factor in delineating the termination of the runaway precipitation process.
Including results from this work, runaway precipitation processes in natural or artificial seawater in comparable studies (Moras
et al, 2022; Hartmann et al., 2023; Fuhr et al., 2022; Pan et al., 2021) approached final $\Omega_{aragonite}$ values between 1.5 and 5.0.
Variations in the final $\Omega_{aragonite}$ in the different approaches might be the result of varying framework conditions during their
runtime.

**4.1.3 $CO_2$-equilibration states**

While the precipitation rates in this study decreased significantly at the end of each experiment, approaching $\Omega_{aragonite}$ values
of 2.5-5.0, it cannot be excluded that further formation of secondary phases could have continued. While some treatments
within the non-$CO_2$-equilibrated experiments still experienced a daily decline of 1-10 µeq kgsw$^{-1}$ in TA during the last 5 days
of operation, these changes were relatively insignificant compared to their earlier rates. Nonetheless, slight changes were still
observed, and it cannot be excluded that the process stopped after a runtime of 25 days.
In contrast to the $CO_2$-non-equilibrated approach, the $CO_2$-equilibrated experiments showed relatively constant final $\Omega_{aragonite}$
values of 5.8-6.0 at the end of the abiotic experiments after 20 days. The runaway process is anticipated to persist at lower
levels of TA loss rates, given that they consistently declined by 20-30 µeq kgsw$^{-1}$ per day during the final 5 days of operation.



### 4.1.4 Comparison to other experiments

Time spans to reach the end of the runaway precipitation process in studies with comparable setups, solely focusing on non-$CO_2$-equilibrated treatments with a $\Delta TA_{added}$ of 2000 µeq kgsw$^{-1}$, ranged from 4 days in Hartmann et al. (2023) to more than 14 days in Moras et al. (2022). In Moras et al. (2022), there were variations in the experimental conditions, such as the use of solid $Ca(OH)_2$ for TA-enhancement, constant agitation, and a temperature of 21°C. These differences may hinder a direct comparison with our study. By contrast, this study employed the configuration introduced by Hartmann et al. (2023), with the only distinction being the utilization of seawater with a salinity of 36.2 and a temperature of around 23°C. In this study, with a salinity of 32.6 and temperatures ranging from 10 to 16°C, the precipitation process in the highest treatments came close to a halt after ~15 days in the non-$CO_2$-equilibrated approaches, while alkalinity was stable over 20 days in $CO_2$-equilibrated treatments with $\Delta TA_{added}$ up to 3600 µeq kgsw$^{-1}$. However, in Hartmann et al. (2023) experiments, significantly faster precipitation rates were observed. This underscores the crucial influence of local environmental factors and application scenario in shaping the dynamics of the runaway process.

Following the $\Omega$-threshold described by Marion et al. (2009), experimental results from Pytkowicz (1973) and considering the general trend predictions from the TA-loss rates (Fig. 5), it is suggested that further treatments in this study might have initiated the runaway process if the experiments had continued. Therefore, treatment levels above $\Delta TA$600 µeq kgsw$^{-1}$ in the non-$CO_2$-equilibrated and $\Delta TA$2400 µeq kgsw$^{-1}$ in $CO_2$-equilibrated approaches had the potential to start carbonate precipitation.

As manipulated water parcels in real world application scenarios would be diluted by untreated water, the results of our study suggest a functional relationship with time for the dilution of TA-enhanced water to non-critical $\Omega_{aragonite}$ values. This could range from minutes to weeks (e.g. see Figs. 4 and 10), dependent on the local physicochemical conditions, the $CO_2$-equilibration state, and the achieved TA levels. Further research is therefore needed to identify the functional relationship for other environmental settings such as temperature, salinity, and the impact of particles for near coastal settings. This is necessary to determine if the here identified relationships are universally valid, or in the context of OAE, further factors need to be considered. In addition, experiments on the dilution of manipulated water masses are needed to test whether TA values exceeding critical ranges can be stabilized. By understanding patterns and factors driving the runaway process, measures could be taken to prevent unwanted consequences during TA addition.

### 4.2 Temporal stability after TA addition

In the context of an open ocean application of OAE, induced turbulence and advective energy in the water would cause the mixing of the alkalized water body with untreated surrounding seawater. Specifically, ship-based applications offer the potential to significantly change the concentrations and saturation states in a relatively short amount of time (e.g. Caserini et



al., 2021; He & Tyka, 2023; Renforth & Henderson, 2017). Dilution could potentially prevent or delay the nucleation phase
for a significant amount of time, to a degree that runaway precipitation events can be avoided at time scales relevant for CDR.
While TA values reached in this study might not represent final targeted TA levels for a real-world application after immediate
dilution with untreated water parcels, studied ranges provide experimental insight into processes during transient enhanced
conditions, occurring around point sources or in (partially-) enclosed water bodies without adequate mixing. Derived from the
results of non-$CO_2$-equilibrated setups, Fig. 10 provides an overview of TA ranges and timeframes until a manipulated water
mass should be diluted to prevent the onset of secondary mineral formation. Note that for the $CO_2$-equilibrated approaches,
this study could not determine reliable comparable stability ranges.

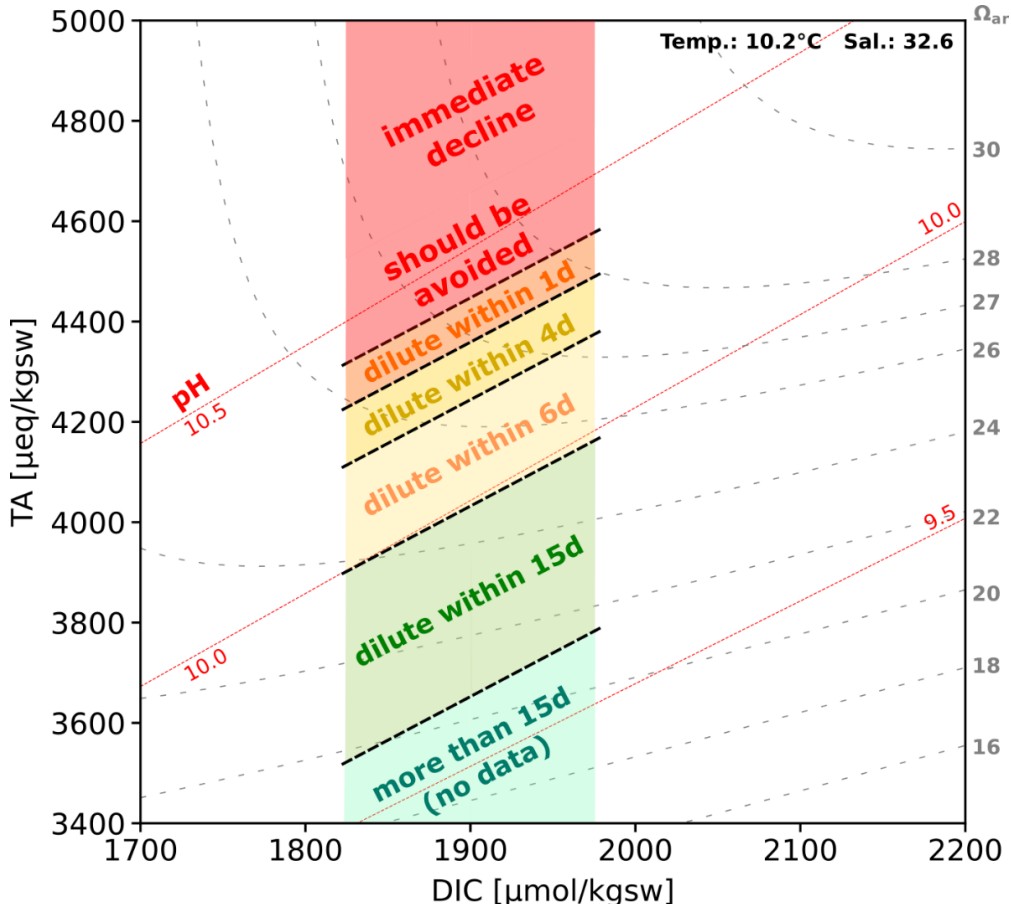

Figure 10: Stability ranges during non-$CO_2$-equilibrated experiments showing upper critical limits for application, which should be avoided to prevent runaway precipitation and its consequences. $\Delta TA0$ (TA: ~2200 µeq kgsw[-1], DIC ~1900 µmol kgsw[-1], pH ~8.2, temperature 10.2°C, salinity 32.6), the red area highlights the critical zone for immediate precipitation, tolerance in time for dilution or other actions to prevent the runaway process are indicated. Ranges in this diagram are applicable only to this setting and should not be generalized.



As described above, treatments which reached TA-levels above 4450 µeq kgsw$^{-1}$ within the non-CO$_2$-equilibrated approach
immediately lost parts of the added TA. To avoid any kind of secondary mineral formation under the local conditions of the
Raunefjord, these values should not be surpassed without additional measures. In theory, every ocean water mass should
possess such a critical threshold level for immediate precipitation. This threshold would determine the practical upper limit for
the application scenarios and overall efficiency of OAE. At present, there does not seem to be a comprehensive method for
calculating this upper threshold. Such a method would need to take into consideration the complexities of the natural
environment such as seasonal and biological cycles, distinct geographical characteristics, and the various approaches of TA
addition. To derive such a method would need systematic research to identify relevant factors and field trials aimed at exploring
their interactions under real-world conditions. Nevertheless, results of this study also showed that TA could be enhanced above
$\Delta TA_{added}$ of 2000 µeq kgsw$^{-1}$ in a non-CO$_2$-equilibrated style without any adverse geochemical consequences if sufficient
mixing is ensured within the given local temporal stability ranges.

## 4.3 Immediate magnesium hydroxide precipitation

If by addition of NaOH to seawater a pH of 10.4 (at 21 °C) is reached, Mg(OH)$_2$ is formed. This is a long-known process as
described by Haas (1916) and Kapp (1928) and comparable to values reached in this study within the non-CO$_2$-equilibrated
treatments, showing an immediate TA decline (pH 10.3-10.6, at 10-12°C). This suggests that the aforementioned process could
potentially account for the immediate precipitation observed in the present experiments. As described by Turek & Gnot (1995),
the formation of Mg(OH)$_2$ during this practice could be considered as an "immediate process", which aligns with observations
in this study. The buffering of TA by Mg(OH)$_2$ formation during TA injection was also supported by the $\Delta TA$: $\Delta DIC$ change
ratios of 2.3-4.9:1 (Fig. 4f) immediately after alkalinity addition. Co-precipitation of carbonate phases besides the generation
of Mg(OH)$_2$ would be necessary to achieve these change ratios and is a regular feature in seawater (Battaglia et al., 2022;
Nguyen Dang et al., 2017; Turek & Gnot, 1995). Formed solid Mg(OH)$_2$ particles could have acted as triggers for further
precipitation of carbonate phases and leading to the observed earlier initiation of the runaway process.
During the injection of NaOH stock solution in seawater, the sharp gradient in pH/TA concentrations could lead to the
generation of lysospheres and floccules, which aggregate and enclose water in its pore space (Turek & Gnot, 1995). Similar
observations were made by Badjatya et al. (2022), who described it as colloidal suspension. While the addition levels were
significantly lower in this study as compared to Badjatya et al. (2022), a comparable trend after the injection was evident. This
trend was visually observable across all non-CO$_2$-equilibrated treatments above $\Delta TA200$. However, to observe the same
phenomenon in the CO$_2$-equilibrated approaches, higher treatment levels (>$\Delta TA2000$) were required to form aggregates.
In the absence of further agitation, aggregates that had formed remained visible for several days. However, when the bottles
were gently rotated after the TA addition, aggregates disintegrated and all treatments that were initially below the critical
threshold for immediate precipitation reached their target TA levels. These observations suggest that the immediate formation



of potential $Mg(OH)_2$ aggregates during the injection process might be reversible, as also noted by Cyronak et al. (2023). In a
real world scenario, wave movement and dilution processes with untreated waters might allow the redissolution of $Mg(OH)_2$
or further metastable carbonate phases.
Effects like an increase in undesired turbidity during TA addition as stated by Eisaman et al. (2023) might therefore be
temporal. Depending on the speed of the redissolution process, sedimentation of the aggregates might function as an export
factor, transferring the added alkalinity to greater depths. The enclosed water in the lysospheres might nevertheless reduce the
sinking rate significantly due to its relatively low density (Turek & Gnot, 1995).





**5 Conclusion**

After the introduction of the runaway precipitation concept by Moras et al. (2022) and Hartmann et al. (2023), this study has identified functional relationships of TA change rates after initiation of the secondary carbonate formation process. With well-defined framework parameters of $\Omega_{aragonite}$, temperature, and salinity, it is therefore possible to predict the temporal evolution of alkalinity. Once the runaway process was triggered, patterns of TA loss were identical for both $CO_2$-equilibrated and non-$CO_2$-equilibrated TA addition approaches. While a progressing runaway process negatively impacts carbonate chemistry parameters, the observed delayed onset of detectable solid phase formation implies that alkalinity could be enhanced beyond 2000 µeq kgsw$^{-1}$, when sufficient dilution with untreated water could be ensured within given time ranges. With knowledge of the local environmental conditions, introduced $\Omega_{aragonite}$, and $CO_2$-equilibration state, it is hypothesized that it is possible to predict a temporary stability range for any given system. The ability to predict the outcomes in advance can facilitate environmental assessments prior to OAE applications. Furthermore, the parameters acquired could be essential for computer models to carry out these evaluations.

Study results are representative for waters with low suspended sediment concentrations, therefore identified TA rate changes during runaway precipitation or temporary stability ranges might differ for systems with higher suspended sediment concentrations, especially if suitable crystal surfaces are abundant (c.f. Moras et al., 2022; Hartmann et al., 2023). Unlike in Hartmann et al. (2023), no relevant differences between biotic and abiotic approaches (distinguished by the filter mesh size) could be identified.

For non-$CO_2$-equilibrated TA additions, an upper pH threshold of around 10.3 could be observed. Crossing this threshold comes with the potential consequence of magnesium hydroxide formation, which was also seen in other studies (c.f. Badjatya et al., 2022; Turek & Gnot, 1995; Vassallo et al., 2021). Considerations about the TA-treatment levels in open ocean application scenarios must therefore consider the onset of $Mg(OH)_2$ formation as an upper threshold. To maximize effectiveness, it is crucial to maintain concentrations just below this critical value when injecting alkalinity into seawater, especially if the local seawater possess efficient dilution capabilities.

These considerations are relevant for modelling the limitations and dynamics of alkalinity enhancement in the ocean, as demonstrated by He and Tyka (2023). Nevertheless, it is essential to validate these findings with in situ experiments to establish parameters and functional relationships applicable to open ocean environments. It is only under these circumstances that accurate assessments can be made. The most promising outcome of this study is the possibility to predict abiotic processes and the stability of alkalinity for effective and realistic applications in the future.



**Author contributions**

The idea for this work was conceived by NS and JH. NS, GF, and CL designed the experiments with help from JH, JS, and UR. NS, CL, GF, and JS carried out sampling and laboratory analysis. NS interpreted the data with help from GF and JH. NS and JH wrote the text with contributions from all co-authors.

**Acknowledgements**

Peggy Bartsch (UHH), Tom Jäppinen (UHH), and Daniel Brüggemann (GEOMAR) are thanked for supporting the preparation and execution of the experiments. This project has received funding from the European Union's Horizon 2020 research and innovation program under grant agreement no. 869357 (project OceanNETs, ocean-based negative emission technologies – analyzing the feasibility, risks, and co-benefits of ocean-based negative emission technologies for stabilizing the climate), as well as the Deutsche Forschungsgemeinschaft (DFG, German Research Foundation) under Germany's Excellence Strategy – EXC 2037 "CLICCS – Climate, Climatic Change, and Society" – Project Number: 390683824, contribution to the Center for Earth System Research and Sustainability (CEN) of Universität Hamburg. Financial support was also provided by the Ocean Alkalinity Enhancement (OAE) R&D Program, a multi-funder effort incubated by Carbon to Sea Initiative via the Ocean Alk-align-project.

**Financial support**

This research has been supported by the Horizon 2020 (OceanNETs (grant no. 869357)), the Deutsche Forschungsgemeinschaft (grant no. 390683824) and the Ocean Alkalinity Enhancement (OAE) R&D Program funded by the Carbon to Sea Initiative.

**Competing interests**

JHA is a co-founder of the Planeteers GmbH. The contact author has declared that all other authors have no competing interests.



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
