# Peer review of "Charly A. Moras"

_EGUsphere, 2023_

## Author Response (AR1)

**Reply to comments on EGUSPHERE-2023-2611**

**Implemented Changes**

**RC1: 'Comment on egusphere-2023-2611', Andrew Dickson, 05 Mar 2024 (https://doi.org/10.5194/egusphere-2023-2611-RC1)**

**I apologize for the time it took me to get to writing this review. I found this to be an interesting manuscript introducing what is likely to become a real problem if Ocean Alkalinity Enhancement (OAE) is to become well established as an mCDR approach.**

Thanks a lot for your dedication. We appreciate the opportunity to refine our work based on your feedback. In the following sections, we have addressed each of your comments comprehensively, aiming to clarify and enhance the quality of our work as per your suggestions.

**The authors carried out a simple series of experiments to assess what happens after increasing seawater alkalinity. They studied the addition of two alternate solutions: one a simple strong alkali, the other a similar solution that had been equilibrated with CO2 at a partial pressure of ~420 µatm. They showed that large additions of alkalinity to a seawater could trigger the precipitation of calcium and magnesium carbonates, as well as magnesium hydroxide, thus reducing the alkalinity in the final solution. If however, the alkaline solution had been pre-equiilibrated with CO2, then higher alkalinity levels could be achieved without triggering the precipitation process. The experiments suggest that the kinetic process that is involved here is somewhat reproducible in its behavior (note: the experiments did not appear to be scrupulously replicated, rather they were repeated with differing initial solution compositions over a range of total alkalinity (and for the addition of pre-equiliibrated solution, also an associated range of total dissolved inorganic carbon).**

**That said, I did find the author's figures and text awkward to read and understand. The difficulty is that they are describing a time-dependent process, where an initial solution composition (formed as a mixture between seawater and a synthetic solution (either NaOH, or an Na2CO3/NaHCO3 mixture with a nomiinal p(CO2) of 420 µatm) changes as a result of precipitating inorganic solids: CaCO2, MgCO3, Mg(OH)2 . The changes are complex due to the equilibrium chemistry of CO2 in such systems, with not only the total alkalinity and total dissolved inorganic carbon changing, but also other compositional properties such as p(CO2), pH, and Ωaragonite.**

**The time-dependent process the authors describe is illustrated as a simple conceptual plot in Fig. 9, and also illustrated as alkalinity loss in Figs. 5 & 6.  Essentially the earlier figures (2,3,4) show the experimental measurements over the course of 20 or 25 days (depending on the experiments). It is these three figures that are hardest to follow (as in them the time dependence is not so clear, and the scales chosen seem somewhat arbitrary).**

Thank you very much, the authors agree that Figures 2, 3 and 4 could be quite complex to read. From the standpoint of the authors Figures 2 and 3 provide a relatively space-efficient option to present the basic parameters. Time-dependent plots like Fig. 6 would double to triple the used space. Also, by omitting the time-dependencies, patterns emerge that might otherwise be difficult to distinguish. For example, the comparison between target and actual reached TA in figs. 2 and 3 demonstrates the

consistency in adjusting the alkalinity to a specific level, as well as the divergence when immediate $Mg(OH)_2$ precipitation prevents further alkalinity increase within the examined ranges. The same applies to the uniform stabilization of TA, $\Omega_{aragonite}$ and pH, which occurred after the precipitation process significantly slowed down. These final configurations adhere to the rules of the underlying carbonate system, provided that the runaway processes are halted at experimentally specific, constant $\Omega_{aragonite}$ values.

Fig. 4 visualizes the dependency of the specified parameters on the carbonate formation (2:1 ΔTA: ΔDIC decline ratio). As one of the main objectives of this publication is to present repeating general patterns, comprehensibly following relatively simple effects of the carbonate/hydroxide formation, the temporal evolution was not the main focus of these diagrams.

The authors would like to provide possible measures to improve their readability:

1)      Additional time-dependent plots will be provided in the supplements

2)      Captions and the related text will be adjusted to improve clarity

3)      Design adjustments in Figs. 3 and 4, particularly for the non-$CO_2$-equilibrated abiotic approach (Fig 3 (d-f) and Fig 4(e)), such as reducing the number of highlighted plots or data points while fading out others. The comprehensive displayed diagrams would then be provided in the supplementary materials.

Time dependent TA, Ωaragonite and pH plots for all approaches were provided in the supplements

(Fig. S5-8)

Number of graphs in Fig. 2, 3 and 4e were reduced, to improve their readability. Comprehensive plots including all datapoints were provided in the supplements (Fig. S2-4)

**Nevertheless, I feel the authors are adequately clear in what they did, and in what they found and I believe the paper is a useful first step in this potentially important area.**

**That said, I have a significant number of small comments (some addressing typos) that I feel the authors should consider changing.**

**The unit "μeq kgsw–1 " for alkalinity seems both old-fashioned, and (sliightly) problematic. First, the use of equivalents has been deprecated in physical chemistry for many decades with moles being a preferred alternative.**

Following your suggestion, "μeq" will be replaced by "μmol" throughout the text.

μeq was replaced by μmol throughout the entire document

**Second, as the experimental solutions are a mixture of natural seawater and another inorganic solution, referring to the amount content of a component as "per kilogram of seawater" seems misleading; strictly it is per kilogram of solution (viz amount content)**

In fact, the experimental solution contained a certain degree of stock solution. For a total amount of 64 treatments, 6 (abiotic $CO_2$-equilibrated, ΔTA 5200-9200) surpassed 1 % of inorganic solution to seawater ratio. While "per kilogram of solution" would be a technically correct description, the authors would like to suggest to use "per kilogram [µmol kg-1]  to minimize the potential for confusion by deviating from the common units used for carbonate chemistry parameters.

per $kgsw^{-1}$ was replaced by per $kg^{-1}$ throughout the entire document

**(line 37) "Once" not "Ones"**

This will be corrected

L42 Typo has been corrected

**(lines 46-48) This does not read right, perhaps words are missiing?**

"Implementation of such identified stability and loss patterns into ocean biogeochemical models, capable of resolving mixing patterns of treated and untreated water parcels, would allow to predict, from the geochemical perspective, safe local application levels of TA, as well as the fate of added alkalinity, and therefore a more realistic carbon storage potential as if neglecting observed carbonate system response to OAE." (L45-48)

Section was rewritten to:

"Incorporating such stability and loss patterns into ocean biogeochemical models, which are capable of resolving dilution processes of treated and untreated water parcels, would, from a geochemical perspective, facilitate the prediction of safe local application levels of OAE. This approach would also allow for an accurate determination of the fate of added alkalinity and a more realistic carbon storage potential estimation compared to the assessments that neglect carbonate system responses to OAE."

L46-50 Changed as stated above

**(lines 58-59) I do not feel that the oceanic residence time for inorganic carbon is a meaningful concept when discussing OAE. If a parcel of seawater is taking up CO2 it is, by definition, at the surface not well-mixed around the ocean.**

Thank you very much for the comment. The authors agree and understand that the sentence was misleading. This will be updated to: "Naturally, inorganic carbon is stored in the ocean over periods of time ranging from 10,000 to 100,000 years (Berner et al., 1983; Mackenzie & Garrels, 1966). This long-term carbon storage potential makes OAE a preferred option over other suggested marine CDR methods."

L62-65 adjusted as stated

**(line 68) "ground" not "grinded"**

This will be adjusted

L73 adjusted

**(Fig. 1 legend). pCO2 is a partial pressure, and thus should be expressed in pressure units (e.g. 420 µatm) not as a mole fraction (420 ppm)**

ppm will be replaced by µatm (Captions Fig. 1 and L106)

ppm was replaced by µatm (Captions Fig. 1 and L110)

**(lines 122-123)  What magnitude are "Minor shifts"?**

In the following table we provide the shifts in the control treatments (ΔTA0), at the start and end (after 25 days) of each experiment:

| Seawater conditions | $CO_2$ state to atmosphere | TA [µmol kg-1] | DIC [µmol kg-1] | pH | $pCO_2$ [µatm] | Ω(ar) | Runtime [days] | Temperature range [°C] |
|---|---|---|---|---|---|---|---|---|
| abiotic | Eq. | -3.96 | 22.10 | -0.050 | 30.44 | -0.25 | 20 | 12-16 |
|  | Non-eq. | -9.24 | 9.11 | -0.080 | 59.32 | -0.14 | 20 | 11-13 |
| biotic | Eq. | -39.97 | 12.03 | -0.120 | 77.08 | -0.50 | 25 | 12-15 |
|  | Non-eq. | -34.39 | -62.20 | 0.024 | -21.32 | 0.27 | 25 | 10-11 |

The table will be added to the supplementary materials and a corresponding reference given in L122.

Table was added to the supplements and a corresponding reference was given in L130

**(lines 125-131) What are the measurement uncertainties of these techniques?**

The uncertainties were set to: TA ±5 µmol kg$^{-1}$; pH ±0.02, based on frequent measurements of standards. The uncertainty for temperature, of ±0.1°C, was taken from the probe's certified specifications. For salinity the uncertainty of ±0.1 was extracted from the calibration standard instructions. These numbers were also used for error propagations in CO2sys based on Orr et al. (2018). This information will be added within section 2.2, "Sampling and measurements"

Uncertainties have been added in L138-139

**(line 195) missing word?**

A comma will be added in L194 and a "such" in L195

Will be adjusted to: "[…], indicating the formation of non-carbonate-bearing secondary phases such as Mg(OH)$_2$ (also see section 4.3)."

To improve the readability the quite long sentence:

Old: "The immediate drop back to 4450 ± 60 µmol kg$^{-1}$ in both non-CO$_2$-equilibrated experiments showed a consistent pattern in dislocation of target and measured TA and DIC values, following a steady declining ΔTA:ΔDIC loss ratio of 4.9-2.3 from highest to lowest target alkalinity levels (Fig. 4f), indicating the formation of non-carbonate-bearing secondary phases, as Mg(OH)$_2$ (also see section 4.3)." was split into two:

New: "The immediate drop back to 4450 ± 60 µmol kg$^{-1}$ in both non-CO$_2$-equilibrated experiments showed a consistent pattern in dislocation of target and measured TA and DIC values. This pattern followed a steady declining ΔTA:ΔDIC loss ratio of 4.9-2.3 from highest to lowest target alkalinity levels (Fig. 4f), indicating the formation of non-carbonate-bearing secondary phases, such as Mg(OH)$_2$ (also see section 4.3)." (L199-202)

A "such" was added in L202

**(line 227) The maximum value for mol% C is high enough to suggest some bicarbonate may also be present in these solids (or there is a typo?)**

Given numbers represent the ranges of nine EDX measurements, which might have caused some confusion, as we didn't exclude the outliers. Ranges will be replaced by the median values: Ca: 8.39 mol%, Mg: 4.07 mol%, Na: 0.65 mol%, C: 20.41 mol%, O: 64.8 mol%, and Cl: 0.47 mol% (Fig. R1). FTIR analysis of the precipitates, as reported in Paul et al. (2024), show an aragonite dominated material. So far, the authors are not aware that bicarbonate phases could be detected to a considerable amount as a precipitated phase during a runaway process as a consequence of OAE.

[Figure]

Fig. R1: Box plot elemental composition of EDX measurements of precipitates, in mol%, n=9

Reference:

Paul, A. J., Haunost, M., Goldenberg, S. U., Hartmann, J., Sánchez, N., Schneider, J., Suitner, N., and Riebesell, U.: Ocean alkalinity enhancement in an open ocean ecosystem: Biogeochemical responses and carbon storage durability, EGUsphere [preprint], https://doi.org/10.5194/egusphere-2024-417, 2024.

L232-233 Ranges were replaced by median values, and the interquartile ranges (IQR) were provided to indicate the variability

**(line 317) What are "varying framework conditions"?**

Factors like: Temperature, salinity, particle load, available reactive surface area, agitation method, $CO_2$-equilibration state, solid or liquid TA addition, type and size of reaction vessel, etc.

For clarification a selection will be given in the text.

Old: "Variations in the final $\Omega_{aragonite}$ in the different approaches might be the result of varying framework conditions during their runtime."

was changed to:

New: "The variations in the final $\Omega_{aragonite}$ across the different approaches could be attributed to differences in framework conditions such as temperature, salinity, $CO_2$-equilibration state, agitation methods or sediment concentration during the course of the experiments." (L323-325)

**(line 326) Is "excluded" the word intended here?**

Will be adjusted to: […] cannot be ruled out that […]

L331 adjusted as stated

**(line 420) Although these experiments had low suspended sediment, I'd hesitate to generalize in this way.**

Will be changed to: Water used in this study had low sediment concentrations, […]

Old: "Study results are representative for waters with low suspended sediment concentrations, […]"

was changed to:

New: "The seawater used in this study had low sediment concentrations, […]" (L425)

**Implemented Changes**

**RC2: 'Comment on egusphere-2023-2611', Anonymous Referee #2, 13 Apr 2024 (https://doi.org/10.5194/egusphere-2023-2611-RC2)**

**Suitner et al. evaluated the tendency of carbonate precipitation associated with ocean alkalinity enhancement. This is a widely interested topic and the paper adds some interesting new experimental results to this thread of research. I have some suggestions for the authors to hopefully help improve the manuscript.**

Thank you very much for your comments. Replies and clarifications to questions and comments are given in detail below.

**First, the authors might want to be more specific and careful in some expressions, for example, in the abstract, Line 27, what does "promising results in the past" exactly mean? You may want to specify to be more "scientific".**

"While various modeling studies showed promising results in the past, […]" will be adjusted to:

"While modeling studies reported a sequestration potential of 3-30 Gt $CO_2$ per year (Oschlies et al., 2023), […]" (L27)

Reference:

Oschlies, A., Bach, L. T., Rickaby, R. E. M., Satterfield, T., Webb, R., and Gattuso, J.-P.: Climate targets, carbon dioxide removal, and the potential role of ocean alkalinity enhancement, in: Guide to Best Practices in Ocean Alkalinity Enhancement Research, edited by: Oschlies, A., Stevenson, A., Bach, L. T., Fennel, K., Rickaby, R. E. M., Satterfield, T., Webb, R., and Gattuso, J.-P., Copernicus Publications, State Planet, 2-oae2023, 1, https://doi.org/10.5194/sp-2-oae2023-1-2023, 2023.

L27 Sentence adjusted as stated above, Oschlies et al., (2023) added to reference list

**Also, in the abstract, concepts like "runaway precipitation" (Line 28-29) and "CO2-equilibrated approaches" (Line 35) are used without definitions, which might be confusing to readers that are not familiar with the field.**

"Recent studies have described the effect of runaway precipitation in the context of OAE, showing that calcium carbonate formation was triggered if $\Omega_{aragonite}$ saturation threshold levels were exceeded." (L28-30) will be adjusted to:

"Recent studies have described the effect of runaway carbonate precipitation in the context of OAE, showing that calcium carbonate formation was triggered if certain $\Omega_{aragonite}$ saturation thresholds were exceeded. This effect could potentially lead to a net loss of the initially added alkalinity, counteracting the whole concept of OAE."

L28-31 Changed as stated. Additionally, the start of the following sentence was adjusted to keep the flow by avoiding starting two consecutive sentences with "This": "**This** precipitation can adversely affect the carbon storage capacity and may in some cases result in $CO_2$ emissions." Changed to: "**The**

**related** precipitation can adversely affect the carbon storage capacity and may in some cases result in $CO_2$ emissions."

"For the $CO_2$-equilibrated approaches, […]" (L35) will be changed to:

"For approaches equilibrated to the $CO_2$ concentrations of the atmosphere, […]"

L36 changed as stated

Also, the authors would like to suggest changing "The $CO_2$-non-equilibrated approaches, […]" (L38) to: "The non-$CO_2$-equilibrated approaches, […]"

L40 and L42 changed as stated

**There is a typo in Line 37, "ones" should be "once".**

Typo will be corrected

L42 Typo has been corrected

**Second, the figures are a little hard to read – y axis are not uniform between "biotic" and "abiotic" conditions (left and right panels in Figure 2 and 3), making it hard to compare the two scenarios.**

y-axes of Figure 2 will be harmonized, the intention behind the non-conformance in axis limits in Fig. 2 was to allow for a comparison with the diagrams in Fig. 3.

Axes of Figure 2 have been harmonized to ensure comparability

**Also, there seems to be way less alkalinity added in the biotic experiments than the abiotic? Why is this?**

The lower amount of added alkalinity within the "biotic" and "abiotic" approaches originates from differing main objectives in the experimental design. The biotic approaches aimed to understand the response of the phytoplankton community. The abiotic experiments were designed to explore stability ranges and precipitation effects, making it essential to expand the examined TA range to much higher levels in the $CO_2$-equilibrated approaches. Within the "abiotic" non-$CO_2$-equilibrated treatments the TA range was increased to confirm the uniform "buffering" effect of $Mg(OH)_2$ formation above ΔTA2400.

**The two terms – "biotic vs. abiotic" are basically "unfiltered vs. filtered". Since the manuscript is basically about thermodynamically-driven inorganic carbonate precipitation, and the two conditions (biotic and abiotic) do not show significant differences (is this true? It is hard to tell with the different TA addition ranges and y-axis), the authors could simply use "unfiltered and filtered seawater" for simplicity.**

Indeed, no significant differences regarding the precipitation behavior could be observed between the "biotic" and "abiotic" approaches (also see L423). Existing deviations most likely originate mainly from temperature differences between the separate experiments.

The wording "biotic" and "abiotic" were used to simplify the comparison with Hartmann et al. (2023) introducing these terms, and to align with the terminology in the upcoming publication on the biological datasets. Both approaches were initially filtered through a 50 μm filter to remove larger biological organisms. The "abiotic" approach was designed to exclude potential influences of biological activity or other factors like natural occurring sediments, achieved by filtering for a second time through a 0.2 μm filter. Therefore, the authors would like to avoid using terms like "unfiltered", as this might be misleading. To enhance comprehensibility, more detailed explanations will be provided in the method section (L107-108).

An additional explanatory sentence was added (L113-115): "The categorization into abiotic and biotic treatments aimed to determine the potential influences of biological activity or naturally occurring sediments, while also preserving the comparability of the experimental setup described in Hartmann et al. (2023)."

**Figure 5 and 6 only have 2 scenarios – abiotic CO2-equilibrated and biotic non-CO2 equilibrated. Why are the other 2 not shown? Biotic CO2-equilibrated and abiotic non-CO2 equilibrated.**

Related diagrams for the other two scenarios are given in the supplements. Taking into consideration the amount of already shown diagrams, the authors could not see a major benefit in showing these in the main body of the text as well.

No changes were made.

**Others:**

Referring to a community comment sent directly to the authors via email, we would also suggest shifting the highlighted ranges by 50 μmol kg-1 towards higher DIC levels to cover a more realistic range of natural seawater.

Figure 10 was updated accordingly. The displayed ranges are now centered around 1960 μmol kg$^{-1}$, as also noted in the caption.